# FREE-MOE: TUNING-FREE MIXTURE-OF-EXPERTS PURIFYING LLMS TO THRIVE ACROSS ANY FIELD

## ABSTRACT

The Mixture-of-Experts (MoE) framework efficiently scales large language models (LLMs) by selectively activating expert subnetworks, reducing computational costs. However, current MoE methods are costly in computation and include additional expert modules that require extra training data for tuning, leading to instability in the optimization process. To address these issues, we introduce FREE-MOE, a tuning-**free MoE** method that leverages pre-trained LLMs' inherent ability to generalize across a wide range of tasks and domains. FREE-MOE dynamically activates experts based on specific domains, achieves improvements while **1) requiring no extra model parameters and 2) being completely tuning-free.** Specifically, we design the DOWP Alg., a **D**omain-**O**riented **W**eight **P**urification Algorithm that purifies the weights in hidden layers and selects the optimal domain-specific experts of domain-specific experts in the hidden layers of the LLM to optimize activation decisions. The activated DSS-Experts, Domain-Specific Subnetwork Experts, can thereby concentrate on specialized task generation, outperforming the corresponding original model. Moreover, FREE-MOE incorporates a multi-level trainable router that activates only the most relevant subnetworks during task, effectively minimizing unnecessary inference computations. Comprehensive evaluations reveals that the DOWP Algorithm consistently achieves general performance gains of 2% to 3%, reaching up to 6.8% across datasets like MMLU, HumanEval, GSM8K, and etc. Additionally, when integrated into FREE-MOE framework, our method demonstrates a cumulative improvement of 1.11% in average. Findings indicate that FREE-MOE not only enhances overall computational efficiency but improves the model's adaptability across any field that encompassed in contemporary language generation model benchmarks, and can be seamlessly applied to any transformer-based LLMs. Code for this project will be released in reachable future.

## 1 INTRODUCTION

The demand for more powerful Large Language Models(LLMs) drives relentless expansion (Kaplan et al., 2020). Yet, the bigger the LLMs, the more they consume: computational resources, time, and energy, which in turn limits their scalability and application efficiency (Patterson et al., 2021; Strubell et al., 2019). Integrating the Mixture-of-Experts (MoE) framework with LLMs has emerged as a highly promising strategy for tackling these challenges, driving significant advancements in the field (Shazeer et al., 2017). Models like Switch Transformer and GShard have demonstrated that model capacity can be scaled to trillions of parameters without a proportional increase in computational costs during inference, setting new benchmarks in various natural language processing tasks (Fedus et al., 2022; Lepikhin et al., 2020). The key innovation of MoE lies in its use of sparse activation, where only a portion of the network is activated for any given input, allowing the model to possess greater capacity without a corresponding increase in computational demand (Zhou et al., 2022).

In a broader sense, the classical MoE approach operates by routing different inputs to specialized subnetwork experts within the model. These expert subnetworks are typically based on diverse foundational models, such as support vector machines (Collobert et al., 2002), Gaussian processes (Tresp, 2000), or hidden Markov models (Jordan & Jacobs, 1994), where the architecture enables the model to specialize in handling a wide range of tasks, improving its capacity to address complex and heterogeneous problem domains. Overall, the pioneering work on MoE introduced mechanisms

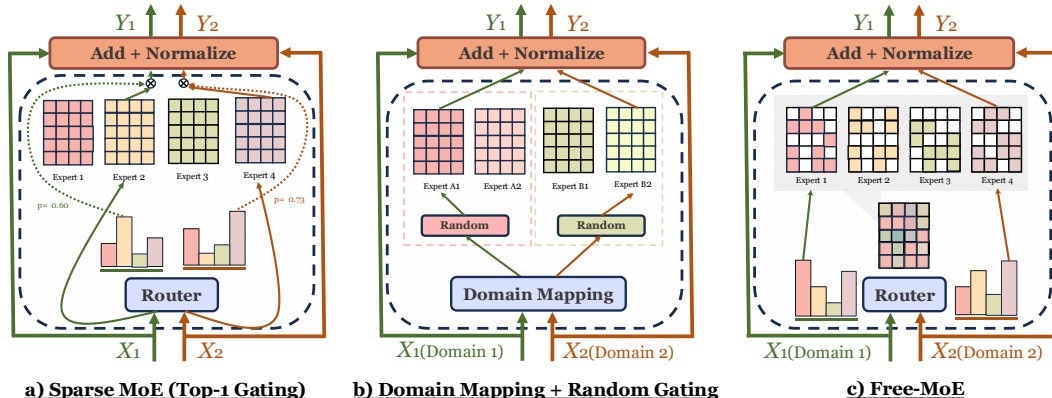

a) Sparse MoE (Top-1 Gating)     b) Domain Mapping + Random Gating     c) Free-MoE

Figure 1: The comparison between traditional MoE methods and our FREE-MOE. (a) represents Sparse MoE with top-1 gating, where the router activates only one expert per input based on the highest probability score. (b) shows the combination of grouped Domain Mapping and Random Gating MoE, which improves task-specific relevance but suffers from inefficiencies due to the random activation of experts. In contrast, (c) illustrates our FREE-MOE, which activates Domain-Specific Subnetwork Experts from pretrained LLMs using the Domain-Oriented Weight Purification Algorithm. FREE-MOE optimizes task-specific responses by purifying out domain-relevant weights to construct experts, achieving a tuning-free effect across diverse tasks.

for dynamically selecting experts, balancing computational load, and addressing challenges such as training instability and communication overhead in distributed systems.

Despite the tremendous potential of the MoE framework, its practical application continues to encounter several significant challenges. A key limitation is that maximizing the benefits of MoE typically requires large-scale, high-quality training data, especially in scenarios where data acquisition is costly or the quality of available data is inconsistent(Lepikhin et al., 2020). Previous work has focused on the field of non-trainable token-choice gating(Roller et al., 2021; Zuo et al., 2022; Gururangan et al., 2022; Ren et al., 2023; Kudugunta et al., 2021), exploring ways to achieve complete load balancing through specific gating mechanisms without requiring additional gating network parameters, thereby improving computational efficiency. However, stepping beyond this perspective, we need to reevaluate how to find more effective tuning-free methods within existing architectures to achieve a higher degree of tuning-free operation. Meanwhile, pre-trained LLMs, which have undergone large-scale training, possess strong capabilities to adapt to various tasks and may offer a new perspective. Therefore, we pose the following question:

*Is it possible to leverage the subnetwork structures within the hidden layers of pre-trained LLMs to enhance task-specific performance and optimize the allocation of computational resources?*

Through deeper investigation, we find that LLMs inherently function as implicit expert networks—while LLMs do not explicitly employ an MoE framework, the subnetworks activated by their hidden layers exhibit specialized behavior across different tasks and inputs. This enables pre-trained LLMs to adapt seamlessly to diverse tasks in tuning-free condition, particularly, without the need for external experts or the addition of significant new parameters. All that is required is an adaptive activation mechanism based on weight purification.

Building on this insight, we propose FREE-MOE, a novel tuning-free mixture of experts architecture designed to leverage the existing subnetwork expert mechanisms within pre-trained LLMs. The key principle of FREE-MOE is its self-directed purification process, where it identifies the input domain and selectively **purifies** the subnetwork weights, retaining only those relevant to the specific task. This allows FREE-MOE to adaptively activate only the expert subnetworks most pertinent to the given task. Compared to Sparse MoE and Domain-Mapping & Random Gating MoE architectures, FREE-MOE achieves improved efficiency and task-specific accuracy without increasing model complexity, thereby significantly reducing the computational burden typically associated with large models, as shown in Figure 1.

Specifically, purification represents the core solution for integrating the MoE framework into LLMs without additional tuning. This process leverages impurity removal and filtration mechanisms, applied during purification, to perform domain-driven expert selection. The core concept involves purifying the hidden layer weights by identifying domain-relevant features from the input, retaining high-contributing expert weights while filtering out irrelevant ones. This approach fully exploits the parameter sharing and hierarchical latent feature structures of pre-trained LLMs, achieving efficient utilization without tuning. In contrast to the simplistic expert selection strategies in Sparse MoE (Fedus et al., 2022) or the full expert activation of Dense MoE (Wu et al., 2022; Dua et al., 2022), purification significantly enhances both the accuracy and effectiveness of expert selection, optimizing model performance for task-specific contexts. Additionally, we define domain-specific expert derived from the hidden layers after purification, explicitly revealing the latent expertise within the pre-trained LLM subnetworks and enabling optimized task-oriented configurations. Compared to the auxiliary load-balancing loss mechanisms in ST-MoE(Zoph et al., 2022), these domain-specific experts offer superior adaptability, facilitating more efficient resource allocation for specialized tasks and substantially improving task-specific model performance. Lastly, we introduce a Trainable Router that dynamically monitors task requirements and domain characteristics, enabling real-time domain identification and efficient allocation of computational resources. Unlike the static mappings employed in Domain Mapping & Random Gating MoE (Ren et al., 2023), our method employs a more granular and dynamic domain activation strategy. This not only enhances operational efficiency but also leads to notable improvements in real-world task performance.

In this work, we present several key innovations that position FREE-MOE as a novel approach to leveraging LLMs within the MoE framework, addressing critical challenges in computational efficiency and task-specific performance:

- **Innovative MoE Architecture FREE-MOE**: FREE-MOE utilizes the latent subnetwork structures within pre-trained LLMs, eliminating the dependency of traditional MoE on large-scale, high-quality training data, and enabling adaptation to multi-task requirements without the need for fine-tuning.
- **Purification Mechanism on Weights**: This mechanism dynamically selects and activates subnetwork weights relevant to specific tasks, significantly improving the accuracy of expert selection. By purifying the hidden layer weights in pre-trained LLMs, the domain expertise of its subnetworks is made explicit.
- **A Multi-Level Trainable Dynamic Router**: We introduce a novel trainable router capable of real-time monitoring of task requirements and domain characteristics, dynamically identifying domains and efficiently allocating computational resources, particularly demonstrating significant advantages in multi-task scenarios.
- **Achieved Significant Performance Improvements Across Multiple Datasets**: Our approach achieved performance gains of 2% to 3% on datasets such as MMLU, MBPP, and GSM8K. Integrated into the FREE-MOE architecture, cumulative improvements of 1.11% validate the effectiveness and practicality of the method.

## 2 RELATED WORKS

### 2.1 THE MIXTURE OF EXPERTS

**The Mixture of Experts (MoE)** introduces multiple specialized expert networks, selectively activating a subset of experts during each inference to maintain model capacity while significantly reducing computational costs (Jacobs et al., 1991; Jordan & Jacobs, 1994; Chen et al., 1999; Tresp, 2000; Rasmussen & Ghahramani, 2001). MoE has demonstrated substantial potential in scaling model size and enhancing performance. Sparse gating MoE layers strike an effective balance between computational cost and model capacity by selecting only a small number of experts for computation per input (Shazeer et al., 2017; Riquelme et al., 2021). The GShard framework by Lepikhin et al. (2020) successfully trained ultra-large-scale models using conditional computation and automatic sharding techniques. Following this, Fedus et al. (2022) introduced the Switch Transformer, which further enhances training and inference efficiency through a simplified expert routing mechanism. To improve expert selection strategies, Zhou et al. (2022) proposed Expert Choice Routing, which optimizes expert selection and load balancing to enhance model performance and resource utilization.

In the realm of multi-task learning, Ma et al. (2018) developed the Multi-Gated Mixture-of-Experts (MMoE) model, which implements task-specific gating mechanisms to capture inter-task relationships. The commonality across MoE methods lies in leveraging sparse activation and expert selection to achieve a trade-off between model capacity and computational efficiency.

## 2.2 INTERPRETABILITY OF LLMS MECHANISM

**Interpretability of LLMs Mechanism** represents a pivotal subfield within the broader study of LLM interpretability, focusing on uncovering the internal mechanisms of language models (Anthropic, 2023; Bricken et al., 2023). The dominant approach conceptualizes LLMs as "circuits", examining neural network hidden representations through feature visualization techniques. This approach has led researchers to identify larger functional components and uncover three key phenomena: Branch Specializatio, Weight Hierarchization and Equivariance (Voss et al., 2021; Petrov et al., 2021; Olah et al., 2020). Recent research has increasingly shifted towards single-layer and multi-layer attention models. In single-layer attention models, structures such as binary tables and skip-trigram tables can be derived from the model's weights. In multi-layer Transformer models, the concept of "induction heads" has emerged—modules formed by the combination of attention heads from different layers, designed to learn patterns within context (Elhage et al., 2021). The core of this research lies in treating neural network components as circuit-like functional modules, while emphasizing that the "circuits" within LLMs are not static but exhibit a high degree of dynamism and adaptability.

## 3 METHOD

### 3.1 DOMAIN-ORIENTED WEIGHT PURIFICATION

The Domain-Oriented Weight Purification, as shown in Figure 2, compresses matrix patches from hidden layers and ranks them by importance. Less relevant patches are purified, reducing complexity while preserving critical weights. This process forms the DSS-Expert for optimized task execution.

**Clustering-based Classification.** Consider multiple datasets $D_A, D_B, \ldots, D_n$, where each dataset corresponds to a distinct domain. These datasets are collectively aggregated to form a comprehensive main knowledge domain set, denoted as $\mathcal{D} = \{D_A, D_B, \ldots, D_n\}$, where each $D_i$ corresponds to a specific domain that encapsulates specialized knowledge relevant to the given tasks. For a given task $\mathcal{T}$, the algorithm identifies the most suitable main knowledge domain $D_j$ by computing the posterior probability $P(\mathcal{D} \mid \mathcal{T})$, which quantifies the likelihood of the task $\mathcal{T}$ belonging to each main domain $\mathcal{D}$. This selection process is expressed as:

$$D_j = \arg \max_{\mathcal{D}} P(\mathcal{D} \mid \mathcal{T}). \tag{1}$$

This ensures that the main domain $D_j$ with the highest probability is chosen, aligning the task $\mathcal{T}$ with the most appropriate domain for further processing. Upon selecting the main knowledge domain $D_j$, the next step involves subdividing this domain into finer subdomains using K-means clustering. For each dataset in $D_j$, feature representations are extracted via the Transformer's embedding layer. The resulting set of feature vectors is denoted as $\mathcal{F}_j = \{F_{j_1}, F_{j_2}, \ldots, F_{j_m}\}$, where each $F_{j_i}$ represents the feature vector corresponding to a data point in $D_j$. The K-means clustering algorithm is then applied to partition the feature set $\mathcal{F}_j$ into $k$ distinct subdomains.

The clustering process aims to minimize the intra-cluster variance, measured by the Euclidean distance between each feature vector and its respective centroid. This optimization is formalized as follows:

$$S^* = \arg \min_S \sum_{i=1}^{k} \sum_{F \in D_{j_i}} \left\| F - \frac{1}{|D_{j_i}|} \sum_{F \in D_{j_i}} F \right\|_2$$

$$= \arg \min_S \sum_{i=1}^{k} \sum_{F \in D_{j_i}} \| F - \mu_i \|_2. \tag{2}$$

By minimizing this objective, the main knowledge domain $D_j$ is effectively partitioned into a set of subdomains, each with a distinct centroid that captures the local structure of the domain. For any

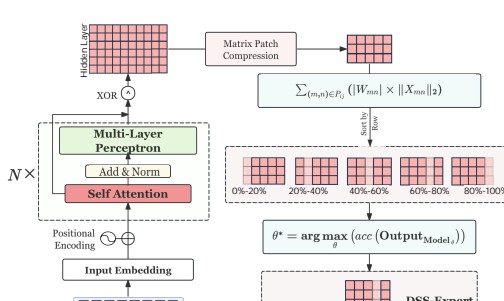 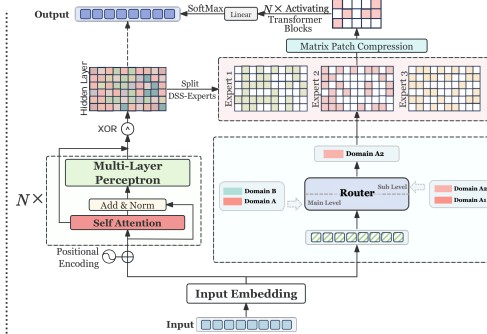

**Step 1. Domain Oriented Weight Purification (1 layer)**   **Step 2. Free-MoE Structure Architecture**

Figure 2: Two-step pipeline for our approach. *Step 1* demonstrates the DOWP (Domain Oriented Weight Purification) mechanism, where hidden layers extracted are compressed, sorted for domain-specific relevance via Equation 4, and further purifying MLP or Self-Attention Layers via Equation 6 to identify and retain the most relevant expert weights. *Step 2* illustrates the FREE-MOE pipeline, in which the router classifies the input by domain, assigning it to its DSS-Expert split from hidden layers, and the output are generated from the activated expert as the task-specific result.

new task $\mathcal{T}$, its feature representation $F_{\mathcal{T}} = Embedding(\mathcal{T})$ is extracted using the Transformer's embedding layer. The task is then assigned to the most relevant subdomain by calculating the Euclidean distance between $F_T$ and the centroid $\mu_k$ of each subdomain. The subdomain that minimizes this distance is selected as follows:

$$k = \arg\min_k \|F_{\mathcal{T}} - \mu_k\|_2. \tag{3}$$

Once the subdomain $D_{j_k}$ is identified, the task $T$ will be assigned to this subdomain. Subsequently, task will be further distributed based on the characteristics of the subdomain, ensuring that the task's features align with the knowledge of the subdomain.

**Metric Calculation on Patches.** After identifying the subdomain $D_{j_k}$, this section is to quantify the critical information contained within it. Activation values serve as an effective metric for assessing the importance of neurons and connections within the network because they directly measure how strongly neurons respond to input features, reflecting their contribution to the network's output (Han et al., 2015). To perform this assessment, we begin by scaling the weight matrix $W$ and feature matrix $X$. A scaling factor $\alpha$ reduces the dimensionality of the matrices, yielding a smaller matrix of size $\beta X \times \beta Y$ (where $\beta = 1/\alpha$). Each element in this reduced matrix corresponds to a patch $P_{ij}$, which represents a subregion of the original matrix and contains condensed information.

In passing, The scaling process ensures that there is no significant loss of information due to the Transformer's self-attention mechanism, which captures global dependencies across the entire input, facilitating information flow between and within patches (see Appendix A).

Afterwards, to further evaluate the significance of each patch, we calculate the following importance metric $\mathcal{M}_{ij}$:

$$\mathcal{M}_{ij} = \sum_{(m,n)\in P_{ij}} \left( |W_{mn}| \times \|X_{mn}\|_2 \right). \tag{4}$$

This metric aggregates the weighted contributions of each element within the patch, thereby quantifying its relative importance to the overall network. The use of both weights and activation values ensures that the local information within each patch is preserved while facilitating efficient global interactions through the self-attention mechanism.

**DSS-Experts Formation.** After calculating the importance metrics $\mathcal{M}_{ij}$ for all patches, these patches are sorted in ascending order based on their importance scores: $\mathcal{M}_{(1)} \le \mathcal{M}_{(2)} \le \cdots \le \mathcal{M}_{(k)}$, where $\mathcal{M}_{(i)}$ denotes the sorted importance scores, and $k$ represents the total number of patches. This sorting process helps identify the least important patches for purification. A threshold range $\theta_r$ is then defined to govern the purification process, specifying the range of importance scores for the patches to be purified:

$$\theta_r = \{\mathcal{M}_{(i)} \mid \mathcal{M}_{(i)} \in [\theta_{\min}, \theta_{\max}]\}. \tag{5}$$

Here, $\theta_{\min}$ and $\theta_{\max}$ represent the lower and upper bounds of the importance scores targeted for purification. Patches within this range are considered less crucial to the model's overall performance and are therefore purified to reduce complexity while maintaining accuracy.

Following the purification process, the model's accuracy is re-evaluated to ensure minimal impact on performance. This accuracy, denoted as $\mathtt{acc}(\theta_r)$, serves to validate the effectiveness of the purification. Subsequently, the outputs from each hidden layer are aggregated, incorporating the purified information to form the DSS-Expert. This approach allows the DSS-Expert to dynamically adapt to domain-specific tasks, ensuring optimized performance.

The final step involves optimizing the purification threshold. By iterating through various threshold ranges $\theta_r$, the model's accuracy is assessed for each range, and the optimal threshold $\theta^*$ is identified:

$$\theta^* = \arg\max_{\theta_r} \left( \mathtt{acc}(\mathtt{Output}_{\mathrm{MODEL}-\theta_r}) \right). \tag{6}$$

This selection of $\theta^*$ ensures the model maintains maximum accuracy while eliminating the least significant patches, resulting in a highly efficient and accurate DSS-Expert, where $\mathcal{E}_{\mathrm{DSS}} = \mathrm{MODEL}_{-\theta^*}$.

---

**Algorithm 1** Pytorch-style pseudocode for DOWP Algorithm.

---

```
# D_k: domain-specific data
# theta_init: initial threshold
# theta_max: maximum threshold
# delta_theta: step size

def optimize_threshold(D_k, Perf, theta_init, theta_max, delta_theta):
    # Initialize variables
    theta = theta_init
    best_perf = 0
    theta_k_star = theta_init

    # Step 1: Extract features and perform clustering
    F_k = embedding(D_k)
    subdomains = kmeans_clustering(F_k, k)

    while theta <= theta_max:
        # Step 2: Calculate importance and select patches for purification
        importance_scores = calculate_importance(subdomains, theta)
        selected_patches = [i for i in range(len(importance_scores))
                                if importance_scores[i] >= theta]

        # Step 3: Purify and evaluate model
        purified_model = purify_model(selected_patches)
        current_perf = Perf(purified_model, D_k)

        # Update best performance and optimal threshold
        if current_perf > best_perf:
            best_perf = current_perf
            theta_k_star = theta

        theta += delta_theta

    return theta_k_star
```

---

`Perf`: function to evaluate model performance on domain-specific data $D_k$.

---

### 3.2 FREE-MOE ARCHITECTURE

The FREE-MOE Architecture, illustrated in Figure 2, utilizes a multi-level trainable router to dynamically classify tasks into main knowledge domains and then subdomains. This routing mechanism activates the most relevant DSS-Experts, ensuring efficient and precise task processing by leveraging purified weights and task-specific parameters.

**A Multi-level Trainable Router.** We introduce a multi-level trainable router that classifies tasks through two hierarchical stages. First, the task embedding $F_{\mathcal{T}}$ is classified into a main knowledge domain $D_j$. Then, in the second stage, the task is further classified into a subdomain $D_{j_k}$ within the selected main domain, as shown:

$$D_{j_k} = \mathcal{R}_{\mathrm{sub},j}(\mathcal{R}_{\mathrm{main}}(F_{\mathcal{T}})). \tag{7}$$

Here, $\mathcal{R}_{\mathrm{main}}(F_{\mathcal{T}})$ maps the task embedding to a specific main domain $D_j$, and $\mathcal{R}_{\mathrm{sub},j}(F_{\mathcal{T}})$, conditioned on this domain, further assigns the task to a subdomain $D_{j_k}$.

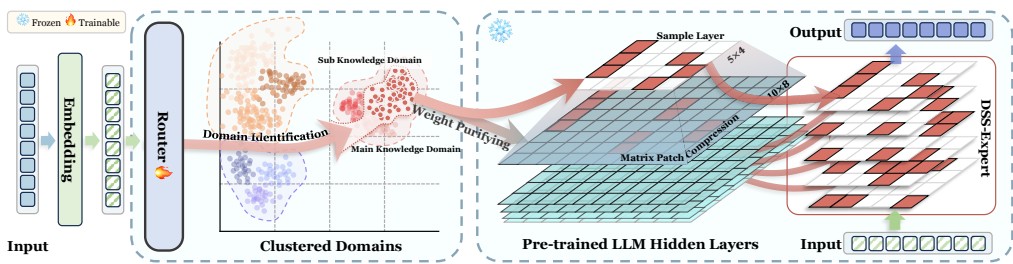

Figure 3: The inference flow through our FREE-MOE. The *input tokens* (⬚) is first embedded and processed by a trainable (🔥) router, where it is initially classified into the main knowledge domain and further into a sub knowledge domain. The *embedded input tokens* (◨) is then passed to the aggregated DSS-Expert, which are dynamically formed from the hidden layers of a frozen (❄️) pre-trained LLM based on the identified domain, then *output tokens* (⬚) are processed by the chosen expert to generate the task-specific result.

The router is trained by minimizing cross-entropy loss at both levels, optimizing the classification process:

$$\mathcal{L}_{\mathcal{R}} = \texttt{CrossEntropyLoss}(X) = -\sum_{k=1}^{m} y_k \log\left(P(D \mid X)\right), \tag{8}$$

where $y_k$ is the true label for domain $D$, and $P(D \mid X)$ represents the predicted probability of the input $X$ being classified into domain $D$.

The multi-level trainable router classifies tasks hierarchically into main and subdomains. The classification process is optimized using cross-entropy loss at both levels, enhancing precision across domains.

**FREE-MOE Inference.** The inference process in FREE-MOE begins with the input tokens $X$, which are first passed through an embedding layer into the embedded representation $\hat{X}$. The embedded tokens $\hat{X}$ are then fed into the router, a trainable classifier designed to assign the input to the most relevant domain. The router $\mathcal{R}$ classifies the input into a main knowledge domain and then sub knowledge domain, depending on the characteristics of the task, expressed as:

$$D_{m_i} = \mathcal{R}(\hat{X}). \tag{9}$$

After classification, the model proceeds `DOWP`. This step filters out irrelevant weights from the pre-trained LLM, retaining only those necessary for the selected domain. The purified weights, denoted as $W_{\text{purified}}$, form the core of the DSS-Expert:

$$W_{\text{purified}} = \texttt{DOWP}(W_{\text{pre-trained}}, D_{m_i}). \tag{10}$$

These purified expert in dynamically activated to process the embedded input $\hat{X}$, ensuring that only the domain-relevant parameters are used for the current task. The embedded input $\hat{X}$ is then forwarded through the selected DSS-Expert, which generate the intermediate output $\hat{Y}$. Finally, the intermediate output $\hat{Y}$ is subjected to a linear transformation and a softmax operation to produce the final output $Y$, which represents the model's prediction for the given task. Whole inference procedure is shown in Figure 2. By dynamically activating only the most relevant experts and focusing on task-specific weights, FREE-MOE optimizes both accuracy and computational efficiency. The process streamlines inference by adapting to the task, ensuring precise and efficient predictions without unnecessary computational overhead.

## 4 EXPERIMENTS

In this section, we evaluate the performance of our DOWP, FREE-MOE, comparing our approach with conventional baseline methods that rely on full network activation, as shown in Figure 4. Our DOWP algorithm consistently boosts performance across datasets: MMLU, MBPP, HumanEval, GSM8K and MathQA. When incorporated into the FREE-MOE architecture, it further improves efficiency, adaptability and stability.

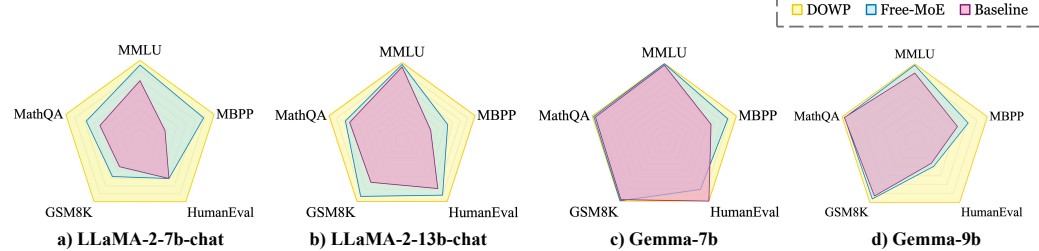

Figure 4: Performance comparison of different models applying DOWP and FREE-MOE. The radar charts illustrate the improvements across five datasets among DOWP ▢ , FREE-MOE ▢ , and baseline methods ▢ .

## 4.1 SETUP

**Datasets.** We select three primary domains for our experiments: **general**, **code**, and **mathematics**. For the general domain, we use the **MMLU** benchmark(Hendrycks et al., 2021), which tests the model's world knowledge and problem-solving skills across diverse subjects. In the code domain, we leverage the **MBPP**(Austin et al., 2021) and **HumanEval** datasets(Chen et al., 2021) to assess coding capabilities, including language comprehension and algorithmic reasoning. For the mathematics domain, we utilize **GSM8K**(Cobbe et al., 2021) and **MathQA**(Amini et al., 2019), which consist of elementary-level mathematical problems covering topics such as algebra and probability. We use the validation sets from **MMLU**, **MBPP**, **MathQA**, and **GSM8K** as reference data for DOWP, with final evaluations conducted on their respective test sets. For **HumanEval**, the MBPP validation set is used as reference data for DOWP, while the evaluation is performed on the full HumanEval dataset. In terms of evaluation setup, we use a **5-shot** approach to assess model performance on the **MMLU** dataset, where the model was provided with five example questions and answers before making predictions. For the other datasets (**MBPP**, **HumanEval**, **GSM8K**, and **MathQA**), we conduct evaluations in a **0-shot** setting.

**Baseline & Foundation Models.** Our experiments utilize four baseline models: **LLaMA-2-7b-chat**, **LLaMA-2-13b-chat**, **Gemma-7b**, and **Gemma-2-9b**. The **LLaMA-2** series and **Gemma** models, based on transformer architecture, are designed for dialogue and natural language understanding tasks. The **7b**, **13b**, and **9b** variants reflect different parameter sizes, balancing computational efficiency with task complexity. Each model consists of stacked layers of **MLP** and **Self-Attention** mechanisms, which are essential for capturing long-range dependencies and processing complex token relationships, making them effective across a range of natural language processing tasks.

**Evaluation Metrics.** We use **accuracy** ($acc\% \uparrow$) as the primary evaluation metric across tasks, defined as the ratio of correctly answered questions to the total number of questions. For **mathematics** and **general** tasks, accuracy is the sole metric for assessing performance. For coding tasks, we evaluate model performance using **pass@1** and **pass@10**.

The **pass@k**($\uparrow$) metric allows the model to generate $k$ different solutions for a given problem and measures the probability that at least one of these $k$ solutions is correct. This is calculated as:

$$\mathbf{pass@k} = 1 - \prod_{i=1}^{n} \left(1 - P_i\right),  \tag{11}$$

where $P_i$ is the probability that the model's $i$-th solution is correct, and $k$ is the number of trials.

## 4.2 MAIN RESULTS.

**DOWP.** We apply the DOWP algorithm to selected pre-trained language models to assess the effectiveness of DOWP in enhancing task-specific accuracy across a range of benchmark datasets. **Firstly,** the application of DOWP results in consistent performance improvements with an ***average gain of 2.04%*** over the baseline models, as shown in Table 1. Specifically, LLaMA-2-7b-chat's accuracy on MMLU increases by 1.98%, while its performance on MBPP (pass@1) rose by 2.08%, and its accuracy on GSM8K improves by 1.97%. Similar patterns are observed for the other models.

Notably, Gemma-7b exhibits an increase of 3.26% in MBPP (pass@10) and a 2.4% improvement in MathQA accuracy, demonstrating DOWP's efficacy across different models and tasks. **Secondly,** as shown in Figure 5, the performance generally lies within *the improvement range of 2% to 3%* across tasks, with the highest reaching up to 6.8%. These gains underscore both the algorithm's effectiveness and the stability of its enhancements. The results suggest that DOWP enhances large-scale models by improving accuracy and maintaining stability. Its ability to purify domain-specific weights ensures efficient operation across diverse datasets and architectures.

Table 1: Performance comparison of DOWP and FREE-MOE with baseline on foundation models.

| Method | | MMLU | MBPP | | HumanEval | | GSM8K | MathQA |
|---|---|---|---|---|---|---|---|---|
| | | acc | pass@1 | pass@10 | pass@1 | pass@10 | acc | acc |
| LLaMA-2-7b-chat | BASELINE | 45.81 | 19.24 | 23.60 | 14.45 | 19.51 | 20.24 | 25.33 |
| | DOWP | 47.79 | 21.32 | 26.40 | 15.73 | 20.73 | 22.21 | 27.34 |
| | Improvement | +1.98 | +2.08 | +2.80 | +1.28 | +1.22 | +1.97 | +2.01 |
| | FREE-MOE | 47.34 | 20.58 | 25.80 | 14.51 | 19.51 | 20.79 | 26.13 |
| | Improvement | +1.53 | +1.34 | +2.20 | +0.06 | ±0 | +0.55 | +0.80 |
| LLaMA-2-13b-chat | BASELINE | 52.34 | 9.68 | 13.00 | 18.66 | 28.05 | 31.77 | 24.86 |
| | DOWP | 53.18 | 11.52 | 16.00 | 19.45 | 29.88 | 34.42 | 27.57 |
| | Improvement | +0.84 | +1.84 | +3.00 | +0.79 | +1.83 | +2.65 | +2.71 |
| | FREE-MOE | 52.93 | 12.39 | 14.80 | 19.13 | 29.27 | 33.86 | 26.34 |
| | Improvement | +0.59 | +2.71 | +1.80 | +0.47 | +1.22 | +2.09 | +1.48 |
| Gemma-7b | BASELINE | 63.56 | 2.94 | 9.00 | 15.31 | 20.12 | 57.92 | 37.12 |
| | DOWP | 65.30 | 6.20 | 15.80 | 16.77 | 22.56 | 59.59 | 39.57 |
| | Improvement | +1.74 | +3.26 | +6.80 | +1.46 | +2.44 | +1.67 | +2.45 |
| | FREE-MOE | 65.05 | 5.85 | 13.90 | 12.93 | 18.29 | 59.29 | 38.79 |
| | Improvement | +1.49 | +2.91 | +4.90 | −2.38 | −1.83 | +1.37 | +1.67 |
| Gemma-2-9b | BASELINE | 69.71 | 8.36 | 9.80 | 12.87 | 18.90 | 68.46 | 50.75 |
| | DOWP | 71.07 | 8.52 | 10.80 | 15.12 | 22.56 | 69.98 | 51.22 |
| | Improvement | +1.36 | +0.16 | +1.00 | +2.25 | +3.66 | +1.52 | +0.47 |
| | FREE-MOE | 70.90 | 8.28 | 10.40 | 14.33 | 20.73 | 69.45 | 50.97 |
| | Improvement | +1.19 | −0.08 | +0.60 | +1.46 | +1.83 | +0.99 | +0.22 |

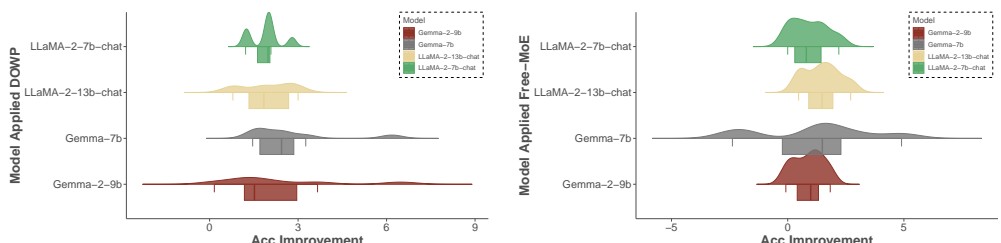

Figure 5: Accuracy improvement comparison of models using DOWP and Free-MoE methods across various architectures.

**FREE-MOE.** To make further comparison, we evaluate the FREE-MOE architecture on the same set of LLMs to examine its effectiveness. **Firstly,** applying FREE-MOE also led to noticeable performance gains of 1.11% in average, though the improvements were generally more moderate compared to DOWP, as shown in Table 1. In detail, LLaMA-2-7b-chat's accuracy on MMLU increases by 1.53%, while its performance on MBPP (pass@1) improves by 1.34%, and its accuracy on GSM8K see a 0.55% rise. Similarly, Gemma-7b's performance in MBPP (pass@10) increases by 2.91%, with MathQA showing a 1.67% gain. These results demonstrate that employing FREE-MOE yields consistent improvements. **Secondly,** the accuracy improvements under FREE-MOE primarily fall within the range of 0.5% to 2.5%, with the highest to 4.9%, as shown in Figure 5.

The results suggest that FREE-MOE, while less impactful than DOWP in terms of absolute gains, offers a viable and stable method for enhancing model performance with minimal additional computation. Its tuning-free nature make it particularly useful for models that need to balance task specificity, suggesting that FREE-MOE could be highly beneficial in scenarios where computational resources are limited, but consistent improvements across a wide range of tasks is still required.

## 4.3 ABLATION STUDIES

In the ablation studies, we use the LLaMA-2-7b-chat model due to its stable performance in previous results. Employing DOWP, we analyze purification ratios, layers, and patch-square configurations. Furthermore, employing FREE-MOE, we analyze the k-means clustering process.

**Purification Ratio.** We purify the all model layers (layers 0 to 31) using a $1 \times 1$ patch-square configuration and examine different ratios on the MMLU dataset categorized into 12 groups. With a **5% purification ratio**, accuracy reaches 47.79%, a 1.98% increase over the 45.81% achieves with a 3% ratio. This demonstrates the 5% ratio effectively balances accuracy and computational cost, as shown in Table 2.

Table 2: DOWP Performance of Different Purification Ratio on MMLU Dataset.

|  | Ratio | MMLU(%) |
|---|---|---|
|  | base | 45.81 |
| Accuracy | 1% | 47.75 |
|  | 3% | 47.85 |
|  | 5% | 47.79 |

**Purification Sublayers.** We then apply a 5% purification to MLP layers and Self-Attention layers, from layers 0 to 31, evaluating the impact on the GSM8K and MathQA, divided into 8 categories. Results shows purifying the Self-Attention layers yields the best on GSM8K, with a 3.18% improvement. On MathQA, the combination performs best, reaching a 2.01% increase, as shown in Table 3.

Table 3: DOWP Performance of Purifying Different Sublayers on GSM8K and MathQA Datasets.

|  | Patch | Datasets | |
|---|---|---|---|
|  |  | GSM8K(%) | MathQA(%) |
|  | base | 20.24 | 25.33 |
| Accuracy | MLP | 21.61 | 26.87 |
|  | Self-Attention | 23.42 | 25.90 |
|  | Combination | 22.21 | 27.34 |

**Patch-square Configuration.** Besides, based on 5% purification ratio, we evaluate different patch-square configurations on the MBPP and HumanEval, divided into 3 categories. For MBPP, the $1 \times 1$ **patch-square** achieves the best performance on pass@1 with a 1.56% improvement, and on pass@10 with a 2.40% increase. Similarly, on HumanEval, the $1 \times 1$ configuration lead with 1.28% gains for pass@1 and 1.22% for pass@10 respectively over the baseline, as shown in Table 4.

Table 4: DOWP Performance of Different Patch Size on MBPP and HumanEval Datasets.

|  | Patch | MBPP(%) | | HumanEval(%) | |
|---|---|---|---|---|---|
|  |  | @1 | @10 | @1 | @10 |
|  | base | 19.24 | 23.60 | 14.45 | 19.51 |
|  | $1 \times 1$ | 20.80 | 26.00 | 15.73 | 20.73 |
| pass@k | $2 \times 2$ | 19.88 | 25.60 | 15.67 | 20.12 |
|  | $4 \times 4$ | 18.58 | 24.00 | 14.88 | 20.73 |
|  | $16 \times 16$ | 18.40 | 24.40 | 13.97 | 17.68 |

**K-means Clustering.** Finally, we examine the impact of different K values applied in FREE-MOE in the K-means clustering step on the MMLU dataset, divided into 12 categories. We vary the number of clusters from 8 to 16, the results show that with **K=12**, the accuracy reaches the highest value of 47.79%, outperforming other cluster settings, as shown in Table 5.

Table 5: FREE-MOE Performance of Different K-means on MMLU Dataset.

|  | K | MMLU(%) |
|---|---|---|
|  | 10 | 47.46 |
| Accuracy | 12 | 47.79 |
|  | 14 | 47.27 |

## 5 CONCLUSION

In this work, we introduced FREE-MOE, a novel framework designed to address tuning-related challenges in the MoE architecture. Specifically, We proposed the DOWP Alg. and incorporated a trainable router to dynamically activate domain-specific subnetworks. FREE-MOE achieves 1) tuning-free, 2) highly portable, and 3) parameter efficiency, and can integrate into any transformer-based LLM without model-specific adjustments. The experiments demonstrated consistent performance improvements, validating FREE-MOE's capability to optimize task-specific responses and its potential for wide application in pre-trained LLMs. Our method thus presents a promising solution for enhancing large model scalability and adaptability.

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

## A    SCALING HIDDEN LAYERS INTO PATCHES MAINTAINS INFORMATION IN SELF-ATTENTION AND MLP

In Transformer models, the Self-Attention and MLP layers are critical for capturing global contextual information and performing non-linear transformations on feature representations. The scaling of hidden layers into patches might raise concerns about the potential loss of information, but this process is designed to preserve both local and global relationships in the model.

**Global Context Preservation in Self-Attention.** The Self-Attention mechanism ensures that every token in the input sequence can attend to all other tokens, capturing global dependencies. This operation is described by:

$$\text{Attention}(Q, K, V) = \text{softmax}\left(\frac{QK^T}{\sqrt{d_k}}\right) V \tag{12}$$

Where: $Q$, $K$, and $V$ are the Query, Key, and Value matrices. $d_k$ is the dimensionality of the Key vectors. By scaling hidden layers into patches, each patch retains local interactions within the patch. Meanwhile, the Self-Attention mechanism ensures that global interactions between patches are maintained. This is because the attention mechanism operates across all patches, allowing the model to propagate global information and maintain context across the entire sequence of patches. As a result, the scaled matrix retains the full global context, ensuring no information is lost during patch scaling.

**Patch Scaling and Information Compression.** When the hidden layers are scaled by a factor $\alpha$, the resulting reduced matrix of size $\beta X \times \beta Y$ ($\beta = 1/\alpha$) has elements that correspond to patches in the original matrix. Each patch $P_{ij}$ captures a compressed representation of the information within the original matrix. By aggregating the contributions from each element in a patch, the scaled matrix effectively compresses the local information, while Self-Attention ensures that this compressed representation continues to interact globally. The importance of each patch is calculated as:

$$\theta_{ij} = \sum_{(m,n) \in P_{ij}} \left( |W_{mn}| \times \|X_{mn}\|_2 \right) \tag{13}$$

This compression allows for efficient representation of both local and global information, preserving the integrity of the original model.

**MLP Layer and Information Flow.** Following Self-Attention, the MLP layer processes the globally-contextualized output. The MLP is defined as:

$$\text{MLP}(h) = \sigma(W_2 \cdot \text{ReLU}(W_1 \cdot h)) \tag{14}$$

Where: $h$ is the output from Self-Attention. $W_1$ and $W_2$ are the weight matrices in the MLP. $\sigma$ is the activation function (typically ReLU). The MLP performs non-linear transformations on the compressed feature representations from the patches. Since the MLP does not rely on spatial relationships, it processes the patch-level information without any risk of information loss. The critical feature transformations in the MLP are unaffected by the scaling process, ensuring that the information flow remains intact.

## B    PROCEDURE OF DOWP TO SELECT THE BEST $\theta$

In this section, we present the procedure for selecting the optimal threshold $\theta$ in the Domain-Oriented Weight Purification (DOWP) method. The goal is to assess the impact of varying $\theta$ values on performance across multiple datasets and domains, specifically MMLU, GSM8K, MathQA, and HumanEval. Each table provides a comprehensive comparison of the DOWP performance over different ranges of $\theta$, from 50% to 100%, highlighting its effectiveness in selecting the most relevant experts in various domains.

Table 6: Performance comparison of DOWP throughout all MMLU domains with different $\theta$.

| cluster_id | Samples | 50-55% | 55-60% | 60-65% | 65-70% | 70-75% | 75-80% | 80-85% | 85-90% | 90-95% | 95-100% | Max (%) | Ratio (%) |
|---|---|---|---|---|---|---|---|---|---|---|---|---|---|
| mmlu_0 | 2047 | 57.79 | 58.72 | 58.96 | 59.26 | 59.99 | 59.94 | 60.67 | 60.23 | 60.77 | 60.92 | 60.92 | 14.58 |
| mmlu_1 | 1518 | 35.57 | 34.72 | 34.32 | 35.70 | 35.38 | 35.31 | 35.84 | 35.44 | 36.17 | 35.77 | 36.17 | 10.81 |
| mmlu_2 | 895 | 23.02 | 24.02 | 22.57 | 23.35 | 23.02 | 22.01 | 22.46 | 22.68 | 22.79 | 22.57 | 24.02 | 6.37 |
| mmlu_3 | 1586 | 48.93 | 47.92 | 49.37 | 48.99 | 49.87 | 49.50 | 49.43 | 50.44 | 50.88 | 51.01 | 51.01 | 11.29 |
| mmlu_4 | 212 | 27.83 | 28.77 | 28.30 | 25.94 | 28.77 | 26.89 | 27.83 | 26.89 | 27.36 | 26.89 | 28.77 | 1.51 |
| mmlu_5 | 1477 | 59.58 | 59.04 | 60.39 | 59.51 | 60.12 | 60.80 | 60.93 | 61.61 | 61.61 | 61.75 | 61.75 | 10.52 |
| mmlu_6 | 322 | 35.09 | 32.92 | 33.85 | 34.78 | 33.54 | 34.78 | 33.54 | 34.16 | 35.09 | 35.09 | 35.09 | 2.29 |
| mmlu_7 | 434 | 27.65 | 25.58 | 29.95 | 26.96 | 26.96 | 27.19 | 28.11 | 29.72 | 27.19 | 29.49 | 29.95 | 3.09 |
| mmlu_8 | 2016 | 55.21 | 55.36 | 55.46 | 55.51 | 56.15 | 55.56 | 56.35 | 57.04 | 57.19 | 57.44 | 57.44 | 14.36 |
| mmlu_9 | 1839 | 46.66 | 47.53 | 46.82 | 46.49 | 47.36 | 47.74 | 47.36 | 48.02 | 48.45 | 48.29 | 48.45 | 13.09 |
| mmlu_10 | 1174 | 30.92 | 30.15 | 31.26 | 31.09 | 30.49 | 30.15 | 30.83 | 32.03 | 32.28 | 31.86 | 32.28 | 8.36 |
| mmlu_11 | 522 | 42.34 | 45.21 | 44.25 | 42.91 | 46.74 | 45.40 | 46.74 | 47.32 | 46.36 | 47.13 | 47.32 | 3.72 |

Table 7: Performance comparison of DOWP throughout all GSM8K domains with different $\theta$.

| cluster_id | Samples | 50-55% | 55-60% | 60-65% | 65-70% | 70-75% | 75-80% | 80-85% | 85-90% | 90-95% | 95-100% | Max (%) | Ratio (%) |
|---|---|---|---|---|---|---|---|---|---|---|---|---|---|
| gsm8k_0 | 240 | 12.92 | 15.00 | 15.83 | 18.75 | 13.75 | 17.08 | 16.67 | 17.50 | 15.83 | 16.67 | 18.75 | 18.20 |
| gsm8k_1 | 8 | 0.00 | 12.50 | 12.50 | 37.50 | 25.00 | 12.50 | 12.50 | 37.50 | 0.00 | 12.50 | 37.50 | 0.61 |
| gsm8k_2 | 225 | 17.78 | 21.78 | 18.22 | 24.00 | 22.67 | 22.67 | 22.22 | 21.78 | 20.44 | 24.00 | 24.00 | 17.06 |
| gsm8k_3 | 361 | 18.28 | 16.90 | 19.67 | 20.50 | 19.94 | 23.82 | 21.05 | 23.82 | 21.61 | 23.82 | 23.82 | 27.37 |
| gsm8k_4 | 113 | 13.27 | 17.70 | 9.73 | 15.04 | 15.93 | 19.47 | 17.70 | 13.27 | 17.70 | 16.81 | 19.47 | 8.57 |
| gsm8k_5 | 193 | 12.44 | 10.88 | 12.95 | 13.99 | 15.03 | 17.62 | 20.73 | 18.65 | 17.10 | 19.69 | 20.73 | 14.63 |
| gsm8k_6 | 6 | 33.33 | 16.67 | 50.00 | 33.33 | 33.33 | 16.67 | 33.33 | 33.33 | 16.67 | 50.00 | 50.00 | 0.45 |
| gsm8k_7 | 173 | 16.18 | 14.45 | 17.92 | 23.12 | 17.34 | 19.08 | 17.34 | 19.65 | 18.50 | 19.65 | 23.12 | 13.12 |

Table 8: Performance comparison of DOWP throughout all MathQA domains with different $\theta$.

| cluster_id | Samples | 50-55% | 55-60% | 60-65% | 65-70% | 70-75% | 75-80% | 80-85% | 85-90% | 90-95% | 95-100% | Max (%) | Ratio (%) |
|---|---|---|---|---|---|---|---|---|---|---|---|---|---|
| mathqa_0 | 289 | 19.03 | 19.72 | 20.42 | 23.53 | 26.99 | 28.72 | 26.30 | 22.84 | 26.30 | 22.15 | 28.72 | 9.68 |
| mathqa_1 | 318 | 24.53 | 24.84 | 24.21 | 22.96 | 31.45 | 23.58 | 29.87 | 29.25 | 26.42 | 25.79 | 31.45 | 10.65 |
| mathqa_2 | 453 | 20.75 | 22.30 | 27.15 | 22.96 | 24.06 | 25.39 | 23.18 | 26.49 | 25.39 | 22.96 | 27.15 | 15.18 |
| mathqa_3 | 107 | 24.30 | 27.10 | 28.97 | 20.56 | 28.04 | 27.10 | 28.04 | 20.56 | 22.43 | 20.56 | 28.97 | 3.58 |
| mathqa_4 | 238 | 24.37 | 20.17 | 29.83 | 21.01 | 22.69 | 29.41 | 22.69 | 27.31 | 29.41 | 28.15 | 29.83 | 7.97 |
| mathqa_5 | 269 | 26.77 | 20.45 | 24.91 | 25.65 | 29.37 | 24.54 | 25.65 | 21.56 | 23.79 | 29.00 | 29.37 | 9.01 |
| mathqa_6 | 659 | 24.28 | 19.58 | 20.49 | 21.40 | 20.64 | 22.91 | 21.55 | 18.97 | 20.64 | 19.88 | 24.28 | 22.08 |
| mathqa_7 | 652 | 20.09 | 21.47 | 21.32 | 24.08 | 22.70 | 22.70 | 25.92 | 24.54 | 23.47 | 22.55 | 25.92 | 21.84 |

Table 9: Performance comparison of DOWP throughout all HumanEval domains with different $\theta$.

| cluster_id | Metric | Samples | 50-55% | 55-60% | 60-65% | 65-70% | 70-75% | 75-80% | 80-85% | 85-90% | 90-95% | 95-100% | Max (%) | Ratio (%) |
|---|---|---|---|---|---|---|---|---|---|---|---|---|---|---|
| humaneval_0 | pass@1 | 44 | 11.82 | 17.05 | 15.68 | 14.77 | 15.00 | 16.36 | 17.05 | 15.23 | 15.45 | 14.77 | 17.05 | 26.83 |
| humaneval_1 | pass@1 | 75 | 8.27 | 9.47 | 10.80 | 13.07 | 10.67 | 15.33 | 12.67 | 13.20 | 13.47 | 13.33 | 15.33 | 45.73 |
| humaneval_2 | pass@1 | 45 | 6.89 | 12.22 | 11.11 | 9.33 | 14.22 | 14.00 | 15.11 | 14.00 | 14.89 | 15.11 | 15.11 | 27.44 |
| humaneval_0 | pass@10 | 44 | 18.18 | 25.00 | 25.00 | 18.18 | 22.73 | 18.18 | 25.00 | 18.18 | 20.45 | 20.45 | 25.00 | 26.83 |
| humaneval_1 | pass@10 | 75 | 10.67 | 14.67 | 13.33 | 12.00 | 18.67 | 18.67 | 16.00 | 17.33 | 17.33 | 16.00 | 18.67 | 45.73 |
| humaneval_2 | pass@10 | 45 | 8.89 | 15.56 | 15.56 | 13.33 | 15.56 | 15.56 | 17.78 | 15.56 | 20.00 | 17.78 | 20.00 | 27.44 |

Table 10: Performance comparison of DOWP throughout all MBPP domains with different $\theta$.

| cluster_id | Metric | Samples | 50-55% | 55-60% | 60-65% | 65-70% | 70-75% | 75-80% | 80-85% | 85-90% | 90-95% | 95-100% | Max (%) | Ratio (%) |
|---|---|---|---|---|---|---|---|---|---|---|---|---|---|---|
| mbpp_0 | pass@1 | 185 | 35.68 | 34.32 | 33.89 | 38.16 | 34.05 | 35.19 | 37.51 | 34.65 | 36.65 | 36.38 | 38.16 | 37.00 |
| mbpp_1 | pass@1 | 53 | 17.74 | 15.47 | 20.19 | 27.92 | 22.83 | 25.47 | 23.96 | 20.75 | 19.62 | 22.08 | 27.92 | 10.60 |
| mbpp_2 | pass@1 | 262 | 7.29 | 6.18 | 5.84 | 6.34 | 6.56 | 8.09 | 6.56 | 6.45 | 7.75 | 7.52 | 8.09 | 52.40 |
| mbpp_0 | pass@10 | 185 | 40.54 | 43.24 | 42.16 | 42.16 | 40.54 | 41.08 | 44.32 | 39.46 | 42.70 | 42.16 | 44.32 | 37.00 |
| mbpp_1 | pass@10 | 53 | 24.53 | 26.42 | 28.30 | 32.08 | 28.30 | 35.85 | 30.19 | 26.42 | 26.42 | 33.96 | 35.85 | 10.60 |
| mbpp_2 | pass@10 | 262 | 9.16 | 9.16 | 9.92 | 9.16 | 10.31 | 11.83 | 9.92 | 10.31 | 11.83 | 11.45 | 11.83 | 52.40 |

## C Threshold-Based Performance Analysis Across Datasets

This appendix provides a comprehensive analysis of the performance trends observed across varying threshold $\theta$ values for the datasets GSM8K, MathQA, HumanEval, MBPP, and MMLU. Each dataset, representing a distinct domain such as mathematics, programming, and general knowledge, showcases unique response patterns when applying the FREE-MOE framework. As $\theta$ increases, we observe noticeable fluctuations in accuracy, highlighting the dynamic behavior of domain-specific subnetworks. The results consistently demonstrate that activating experts based on purified domain-specific weights yields stable improvements across tasks, while maintaining computational efficiency. This analysis reinforces the scalability and adaptability of FREE-MOE, validating its ability to enhance task-specific accuracy without the need for fine-tuning.

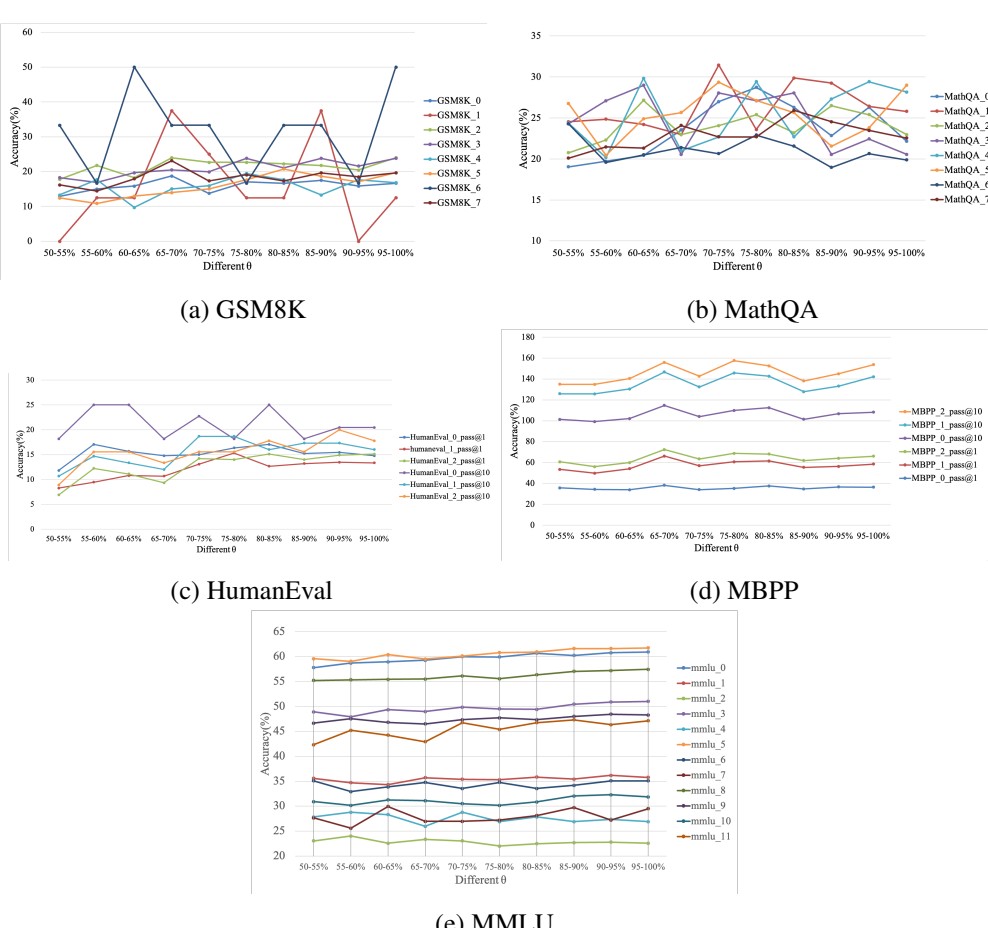

Figure 6: Accuracy comparison of DOWP across different thresholds $\theta$ for various datasets including GSM8K, MathQA, HumanEval, MBPP, and MMLU. Each subfigure (a-e) shows performance variations with respect to the $\theta$ values, highlighting dataset-specific accuracy trends.

