# OpenReview forum: "Free-MoE: Tuning-Free Mixture-of-Experts Purifying LLMs to Thrive across Any Field"
_ICLR.cc/2025/Conference — Submitted to ICLR 2025_

### Official Review · Reviewer_fu2A · 2024-10-26

**Soundness:** 2
**Presentation:** 1
**Contribution:** 2
**Rating:** 3
**Confidence:** 4

**Summary:**

This article introduces a method to reduce inference costs and improve large language model (LLM) performance by activating only a subset of its parameters based on task-specific data. The method first classifies the input embedding vector into a sub-domain, then refines the relevant parameters to create a domain-specific subnetwork using the Domain-Oriented Weight Purification (DOWP) algorithm. This tailored subnetwork processes the input embedding vector to generate the final output.

**Strengths:**

- Enhance LLM performance on specific tasks while simultaneously reducing inference costs.
- Introduce a new LLM architecture that leverages the Mixture-of-Experts concept.

**Weaknesses:**

- The clarity of the proposed method's description could be improved; please refer to the questions section for details.
- The claims lack accuracy. For instance, the authors describe the method as "tuning-free," yet it still requires "training" for the router.
- The notations are inconsistent in Sections 3.1 and 3.2:
  - The interchangeable use of "domain" and "dataset" in Section 3.1 causes confusion, making it unclear whether $n$ refers to the number of datasets or domains. Additionally, it is unclear whether $D_A$ in the third paragraph of Section 3.1 refers to a domain or a dataset. Furthermore, the meaning behind $S$ and $D_{j_i}$ remains undefined.
  - The definition of $Y$ presented in Line 250 is missing.
  - The notation $k$ has multiple meanings in the paper. it denotes the number of patches in Line 265 but represents the number of clusters in previous paragraphs. Furthermore, in Algorithm 1, $k$ appears in `kmeans_clustering` without prior definition.
  - In Section 3.1, it is unclear if "feature matrix $X$" refers to the activation embedding vector or the self-attention head.
  - In Section 3.2, $X$ represents input tokens, while in the previous section, $X$ is used to denote the feature matrix.
-  Several typos appear in the paper:
   - In the first paragraph of page 5, “the Euclidean distance between $F_{\mathcal{T}}$” not $F_T$.
   - In the 3rd paragraph of page 5, “In passing, **the** scaling”, not “The”.
   - In the 5th paragraph of page 7, “These purified **experts** **are** dynamically activated”, not “expert in”.
   - In the 5th paragraph of page 7, “Whole inference procedure is shown in **Figure 3**.” It should not be “2” based on the context.

**Questions:**

- In the abstract and introduction sections, the authors report performance gains of 2% to 3% and 1.11%. Could they clarify which model these improvements are based on?
- Could the authors explain which MoE addresses communication overhead in distributed systems in Lines 76-78? I find it difficult to see the connection between the first part of this paragraph, which discusses various expert foundation models.
- Could the authors explain the difference between “main knowledge domain” and “domain” mentioned in Lines 192-195?
- Since the authors claim that Free-MoE enhances overall computational efficiency, could the authors provide an inference time comparison?
- Could authors clarify the difference between DOWP and Free-MoE?
- Could the authors elaborate on the rationale behind classifying a domain into sub-domains? The reviewer thinks the motivation is unclear.
- Please address the following questions regarding Fig. 2:
    - In Step 1, why is there an “XOR” operator before the hidden layer?
    - In Step 1, what does “sort by row” mean? Is it relevant to the sorting process of importance score $\mathcal{M}_{ij}$?
    - In Step 2, why is there a dashed arrow from the hidden layer to the output? It seems contradictory to Fig. 3, where the output comes from the path through DSS-Expert only.
- The pseudo-code and figure expression seem inconsistent. In Step 2 of Algorithm 1, it seems that the patches with importance scores greater than $\theta$ are taken. However, Fig. 2 does not align with this algorithm. Could the authors explain this clearly?
- The details regarding the K-means algorithm are missing:
  - Did the authors use one token or one batch for each domain in their experiments?
  - Could the authors explain the initialization strategy used in the K-means algorithm?
- In inference mode, when forming the DSS expert, the authors mention in Fig. 3 that the DSS experts are dynamically formed. Could the authors clarify whether this requires iteratively finding the best threshold for each token or for each batch?
- After performing K-means clustering and finding each sub-domain centroid, should this centroid be cached and reused when inferencing?
- The author only conducts experiments on the datasets that are already used to train routers and perform K-means clustering. Can this method be applied to other datasets that also fall in the primary domain (e.g., “general”, “code”, “math”)?
- What is the relationship between this work and “Interpretability of LLMs Mechanism” mentioned in Section 2.2?

**Writing Suggestions**
- The reviewer suggests changing the term “domain-specific experts” in the abstract to “the optimal experts among domain-specific experts” for enhanced clarity.
- The reviewer suggests removing redundant sentences:
  - In Lines 241-243, the two sentences “the task T will be assigned to this subdomain” and “Subsequently, task will be further distributed based on the characteristics of the subdomain” seem to have the same meaning.
  - In Line 267, ”This sorting process helps identify the least important patches for purification.” seems redundant.

---

### Official Review · Reviewer_GaFy · 2024-11-01

**Soundness:** 3
**Presentation:** 3
**Contribution:** 3
**Rating:** 5
**Confidence:** 3

**Summary:**

This paper introduces Free-MoE, a tuning-free mixture-of-experts (MoE) method that harnesses the inherent generalization abilities of pre-trained large language models (LLMs) to adapt across diverse tasks.

**Strengths:**

(1) The method seems to be interesting and novel, as it effectively leverages the inherent MoE structure from pre-trained dense LLMs. This approach intuitively simplifies MoE fine-tuning and training while stabilizing the optimization process.

(2) Extensive experiments are conducted across various models, demonstrating the method’s effectiveness and scalability.

**Weaknesses:**

(1) While the motivation and main idea are clear, the implementation details in Algorithm 1 are somewhat confusing:
      (a) What does Perf mean? It's unclear whether Perf is a specific performance metric or an abstract indicator for evaluating expert effectiveness.
     (b) How is the weight matrix scaled by alpha? Is this scaling achieved through random search, or are heuristic methods applied to determine alpha?
    (c) What is the meaning of Patch? The reference to an element in the reduced matrix as a patch adds to the confusion—could this term imply a specific structure or operation within the matrix?
(2) The paper doesn’t explicitly discuss efficiency drawbacks. From my understanding, the method seems to rely on maintaining indices to store expert-related information across coarse to fine-grained features, potentially causing efficiency degradation during inference.
(3) It's not immediately clear why performance would improve with only DOWP, as fewer parameters are utilized during inference, which would typically suggest a decrease rather than an improvement in performance.

**Questions:**

see weakness

---

### Official Review · Reviewer_e9HR · 2024-11-04

**Soundness:** 2
**Presentation:** 1
**Contribution:** 1
**Rating:** 3
**Confidence:** 5

**Summary:**

This paper proposes a tuning-free method to convert a non-MoE LLM into a MoE LLM to reduce computational overhead. Specifically, for each linear layer, Free-MoE uses a calibration dataset containing N tasks to generate M task-specific **sparse weight matrices** as experts based on the original **dense matrix**. The authors then train a multi-level router that first selects the set of task-specific experts for each input sample and subsequently identifies the most relevant experts within that set. Experiments show that on models smaller than 13B, Free-MoE can improve performance on specific tasks.

**Strengths:**

1. On more challenging datasets such as MMLU, GSM8K, and MathQA, Free-MoE demonstrates good algorithmic performance, showing that this method has the potential for further exploration.

**Weaknesses:**

1. The writing of this paper needs to be checked very carefully, especially the logic and terminology. For example:

    - Logic: The logic of abstract is hard to follow. In the first sentence, the authors mentioned that the MoE canreduce the computation cost. However, in the second sentence, the authors mentioned MoE methods are costly in computation. The statements are quite conflict.

    - Logic: Again in the abstract. In the second sentence, the authors mentioned that the current MoE methods have three issues require extra training data which can lead to instability in the optimization process. However, why more training data can lead to instability in training? It seems that it is not necessary to mention this issue, because the Free-MoE does not solve the instability issue.

    - Terminology: In line 21, the authors define "domain-specific experts" and in line 23, the authors define "Domain-Specific Subnetwork Experts". Is there any difference between these two terms?

2. The equations in this paper need detailed explaination. The equations are hard to understand now.

    - For equation 1, how can we solve this estimation regarding the datasets and tasks?

    - For equation 2, the embedding of each sentence is a matrix with different sizes. How to convert the embedding matrix into embedding vector?

    - For equation 4, the size of weight matrix and the input activation are always different. Why can we use the same index to select numbers in both matrices? What does P_{i,j} stand for?

3. Larger models are more likely to encounter excessive computational demands, yet the authors only conducted experiments on models below 13B, lacking tests on larger models like LLaMA3-70B and Qwen2-72B.

4. Some tiny writing issues:

    - In line 21, "selects the optimal domain-specific experts of domain-specific experts in the hidden layers", it seems that the authors repeat the "domain-specific experts".

**Questions:**

1. How many experts will be generated by the proposed method for each task?

  2. What is the density of each generated expert on different tasks? Are there any new findings? For example, do math datasets need more dense weight than other datasets, such as law datasets in MMLU?

---

### Official Review · Reviewer_fZP1 · 2024-11-04

**Soundness:** 2
**Presentation:** 1
**Contribution:** 2
**Rating:** 3
**Confidence:** 4

**Summary:**

The paper introduces Free-MoE to enhance the efficiency and adaptability of non-MoE LLMs across various tasks and domains without requiring additional tuning. Free-MOE propose a Domain-Oriented Weight Purification (DOWP) Algorithm to generate domain specific experts for each linear layer. Then Free-MoE propose a multi-level router to select optimal experts for each input instance during inference. This approach achieves performance improvements of 2% to 3% across multiple datasets without tuning.

**Strengths:**

The authors conducted detailed experiments to verify the effectiveness of their algorithm and provided a comprehensive ablation study. The results indicate that Free-MoE can further improve model performance on specific tasks.

**Weaknesses:**

1. Lacking specific implementation details. Including: How many experts will Free-MoE produce in total? Does storing these experts require additional memory overhead, and if so, how large is the memory cost?

  2. A more comprehensive evaluation of the approach is lacking. While the experts identified for one specific task may improve performance on that specific task, how much does this affect performance on other tasks? Is the performance drop on other tasks acceptable?

  3. There is no hardware performance evaluation and analysis. It seems that each expert generated by Free-MoE has an unstructured sparse pattern. In this case, it is hard to accelerate the inference speed, and a special system design is needed to reduce the computation overhead. For each task, can you provide the inference speed and summarize the FLOPS results in the experiment part?

**Questions:**

What is the difference between the proposed Free-MoE and unstructured weight pruning? I recommended the authors add the discussion in the related work.

---

### Meta-Review · Area_Chair_XvTD · 2024-12-19

**Metareview:**

This paper introduces an approach to identify experts from pre-trained non-expert LLMs via a weight “purification” strategy that is similar to existing weight pruning strategies akin to identifying lottery ticket subnetworks for different tasks. The paper focused on pretrained LLMs and presents a method that does not require any tuning of LLM weights. Experts are selected for each linear layer based on identifying tasks/subtasks via a K-means clustering approach applied to the Transformer embedding layer. Neurons with a pretrained model (LLM) are selected per expert by choosing those that have large activations to examples within a task, while neurons with small activations are pruned. A router is trained via CE on Transformer features to identify the task/sub-task cluster corresponding to a given example. The weight purification approach and the full Free-MoE model are compared to existing benchmarks on a set of tasks corresponding to language understanding, code generation, and math reasoning. Experiments indicate that the proposed approach improves consistently over baseline LLMs.

Strengths:
The approach appears novel and may have practical value as a way to improve inference efficiency and task accuracy in some LLM settings without expensive fine-tuning and avoiding optimization instability issues with MoEs in general. The authors validate the approach with well-designed ablations. The method demonstrates consistent improvements over baselines on challenging benchmarks.

Weaknesses:
The model derivation is difficult to follow and lacks implementation details (including information about the number of experts generated for each task, memory overhead, etc., making it not easy to reproduce or verify. Analysis is lacking in terms of how the underlying pruning method differs from standard weight pruning and why it would improve over baselines. In addition, the paper lacks an analysis of inference time efficiency improvements, which is a core motivation for the approach. Further experiments would be helpful that decoupling the data used to train the routers/clustering from inference.

Decision reasoning:
While the method is interesting and has potential for being a practical contribution, the weaknesses currently outweigh the strengths. Reviewers were unanimous in expressing concern over method and experimental details and all recommended rejection and focusing on improvements. In addition, the authors do not appear to have made any effort to address the reviewer feedback.

**Additional Comments On Reviewer Discussion:**

The reviewers all expressed similar concerns about the method and experiment incompleness. Unfortunately, the authors never took the opportunity to respond to reviewer feedback, and my own reading agrees with the assessment of the reviewers about the quality of the paper.

---

### Decision · Program_Chairs · 2025-01-22

Reject